# HashAttention: Semantic Sparsity for Faster Inference

**Aditya Desai** [1]  **Shuo Yang** [1]  **Alejandro Cuadron** [2]  **Matei Zaharia** [1]  **Joseph E. Gonzalez** [1]  **Ion Stoica** [1]

## Abstract

Leveraging long contexts is crucial for advanced AI systems, but attention computation poses a scalability challenge. While scaled dot-product attention (SDPA) exhibits token sparsity, i.e. only a few pivotal tokens significantly contribute to output, exploiting this sparsity remains challenging. Existing methods either suffer from quality degradation or require substantial additional resources. We show that identifying pivotal tokens is a Maximum Inner Product Search (MIPS) problem. However, existing MIPS solutions are not well-suited for SDPA, as they are not GPU-friendly and often underperform due to the separated query and key distributions. This paper introduces HashAttention, framing pivotal token identification as a recommendation problem. Given a query, HashAttention encodes keys and queries in Hamming space, capturing the required semantic similarity, using learned mapping functions. HashAttention efficiently identifies pivotal tokens for a given query using bitwise operations and computes attention using only these tokens, improving the overall attention efficiency. Trained on generic data, HashAttention reduces tokens used by up to $16\times$ with minimal quality loss, requiring only 32 bits of auxiliary memory per token. Sparsity can be further improved to $32\times$ through task-specific fine-tuning. On A100 GPU, at $32\times$ sparsity, incorporating HashAttention reduces attention latency by up to $4.3\times$ in GPT-FAST and $2.54\times$ in FlashDecode, and achieves up to $3.12\times$ higher throughput for GPT-FAST.

## 1. Introduction

The ability to refer to long contexts efficiently is crucial for modern AI applications, from processing lengthy documents

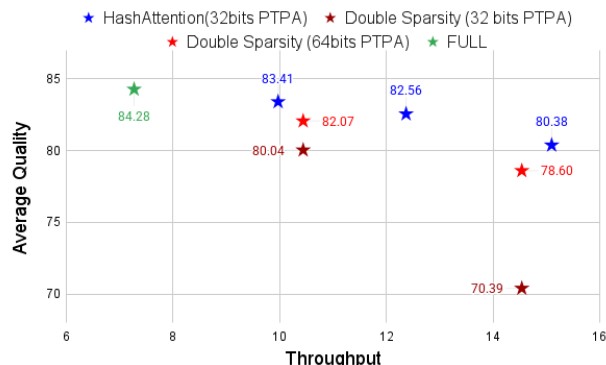

*Figure 1.* Quality vs. Throughput (on A100 GPU) vs. Auxiliary memory at different sparsity rates for Double Sparsity (16 channels, 2bit and 4 bit quantization) and HashAttention implemented in GPT-FAST (batch=1) framework on RULER@32K using Llama-3.1-8B-Instruct. The impact of auxiliary memory is excluded from throughput computation due to the relatively small scale of the problem (batch=1 and 32K context length). Also, throughput is overestimated for Double sparsity by excluding quantization/dequantization in throughput computation.

to engaging in extended conversations (Touvron et al., 2023; Achiam et al., 2023; Liu et al., 2024b). It is commonplace for LLM models to preprocess and store huge amounts of text in the form of KV Cache, which is later used to process various prompts. However, the Scaled Dot Product Attention (SDPA), which is fundamental to the transformer architecture that has driven the Generative AI revolution (Vaswani, 2017; Brown et al., 2020), does not scale well with context length. Generating a single token requires SDPA to access the entire context KV Cache, which can be hundreds of GB. For instance, a KV Cache of 512K tokens is 64GB (0.125 MB per token) for the LLama3.1-8B model.

Luckily, sparsity naturally arises in SDPA. Due to the softmax kernel, only a few *pivotal* tokens significantly contribute to the final attention computation(Bricken & Pehlevan, 2021; Deng et al., 2024). Efficiently identifying these pivotal tokens provides a pathway to achieving efficient attention. Various approaches to identifying pivotal tokens have been explored in the literature. Heuristic-based methods, such as fixed sparsity patterns(Xiao et al., 2023), ignore the dynamic nature of contextual sparsity in attention, resulting in suboptimal attention quality. KV cache discard strate-

[1] Department of Electrical Engineering and Computer Sciences, UC Berkeley [2] Department of Computer Sciences, ETH Zurich. Correspondence to: Aditya Desai <apdesai@berkeley.edu>.

*Proceedings of the 42nd International Conference on Machine Learning*, Vancouver, Canada. PMLR 267, 2025. Copyright 2025 by the author(s).

gies, such as those proposed in (Liu et al., 2024c; Zhang et al., 2023; Li et al., 2024a), identify the global importance of tokens per attention head, incorporating some degree of dynamic sparsity. These methods discard tokens based on their observed importance in context prefixes. However, token importance is dynamic, and tokens deemed unimportant in prefixes can become critical for future inference, leading these methods to fail in certain scenarios (Xiao et al., 2024; Tang et al., 2024). Some approaches to truly dynamic sparsity have emerged(Xiao et al., 2024; Yang et al., 2024; Tang et al., 2024). However, firstly these methods rely on heuristics to reduce the computation of pivotal tokens, leading to low recall rates. Secondly, improving recall with these methods needs substantial auxiliary memory.

In this paper, we take a principled approach to identifying pivotal tokens. Given a query and a large set of key-value pairs, we frame the task of retrieving important tokens as a recommendation problem (Zhang et al., 2021). We show that identifying pivotal tokens under the independence of value vectors is exactly a Maximum Inner Product Search(MIPS) problem. Existing solutions to MIPS are unsuitable for pivotal token identification due to two reasons (1) SOTA solutions to MIPS are based on graph data structures which are GPU unfriendly. (2) Query and Key distributions are separated in Attention causing off-the-shelf solutions to fail (Liu et al., 2024a). We observe that MIPS can be reduced to a cosine-similarity search (Neyshabur & Srebro, 2015; Shrivastava & Li, 2014) which can be further approximated using bit signatures (Gionis et al., 1999), or embeddings in Hamming space.

HashAttention uses learned independent mappings to embed the key-value pairs and queries in a Hamming space. These mappings are trained on the actual key-value and query data from the LLM model to encode inner product-based similarity into bit signatures. Given a query, the tokens with key-value signatures closest to the query signature in Hamming space are identified as pivotal tokens. These signatures can be packed in integers enabling their efficient storage and processing. HashAttention is GPU friendly and resolves the issue of separated distribution of queries and keys using learned independent mappings.

An instance of throughput improvement for GPT-FAST, on RULER@32K benchmark with Llama-3.18B-Instruct is shown in Figure 1. At 32K context length, HashAttention improves the throughput $1.3\times$ ($4\times$ sparsity) at quality drop of 0.8 points and $1.7\times$ ($8\times$ sparsity) with quality drop of 1.7 points using auxiliary memory of 32 bits per token per attention head (PTPA). At auxiliary memory of even 64 bits PTPA, Double Sparsity(DS), a SOTA sparse attention, fails to preserve model quality and improve throughput. Broader experiments reveal more potential for sparsity and throughput improvements.

HashAttention, trained with a generic dataset (i.e. independent of downstream tasks) can give up to $16\times$ reduction in KV cache usage while preserving the model quality. For instance, with $16\times$ sparsity, HashAttention maintains model quality within a drop of 0.78 points on LongBench(Bai et al., 2024b) and 1.13 points on RULER@16K using Llama-3.1-8B-Instruct model. Our ablations show that sparsity can be further improved if HashAttention is finetuned on task-specific data. For instance, on some Longbench datasets with Llama-3.1-8B-Instruct and Mistral-7B-v.03-Instruct, we can go up to $32\times$ sparsity without loss of model quality.

Incorporating HashAttention in inference systems can improve the latency and throughput of inference. For instance, HashAttention-32$\times$ improves latency of GPT-FAST(GPTFast, 2024) attention upto $4.3\times$ and that of FlashDecode(Dao, 2023) upto $2.54\times$. Additionally, end-to-end model throughput for GPT-FAST, tested with Llama3.1-8B model, can be improved up to $3.12\times$ with HashAttention-32$\times$. Furthermore, HashAttention only uses 32 bits PTPA auxiliary memory. To achieve the same quality, other sparse-attentions either need around $4\times$ more pivotal tokens (i.e. $4\times$ less sparsity) at 32 PTPA or need more auxiliary memory to operate at the same sparsity. The code for HashAttention is here. [1]

## 2. Related Work

### 2.1. Long Context and Retrieval-Augmented Generation

Recently, there has been rising interest in developing models capable of processing long contexts(Beltagy et al., 2020; Guo et al., 2022; Han et al., 2023; An et al., 2024; Ma et al., 2024), driven by emerging applications such as multi-document question answering, multi-turn chatbots, and tutor bots (Bai et al., 2024a; Feng et al., 2022; Davies & Murff, 2024). Two main approaches to addressing this challenge are: (1) designing models that natively support long contexts using methods like ROPE scaling (Su et al., 2023), and (2) employing a retriever to extract relevant portions of text to augment the prompt, a technique known as Retrieval-Augmented Generation (RAG) (Lewis et al., 2020).

A recent study highlights the advantages of building long-context LLMs over RAG approaches (Li et al., 2024b). Notable progress in this area includes the emergence of long-context models, such as the open-source Llama 3.1 (Dubey et al., 2024) series, which supports a context length of 128K tokens, and proprietary models like GPT-4 Turbo (Achiam et al., 2023), Claude-2, and Gemini 1.5 (Team et al., 2023), offering context lengths of up to 128K, 200K, and 1M tokens, respectively.

---

[1]https://github.com/xAlg-ai/HashAttention-1.0

## 2.2. Post-training Sparse Attention

We classify the previously proposed approaches to sparsify the attention computation of pre-trained models as follows:

**Fixed Sparsity:** Approaches such as StreamingLLM (Xiao et al., 2023) adopt a fixed sparsity pattern to approximate attention based on the observed importance of attention sinks (e.g., the first few tokens) and recent tokens. However, subsequent studies (Zhang et al., 2023; Xiao et al., 2024) have demonstrated the dynamic nature of sparsity and the resulting quality degradation when using fixed sparsity.

**KV Cache Discarding:** Methods such as H2O(Zhang et al., 2023), ScissorHands(Liu et al., 2024c), FastGen(Ge et al., 2023), and SnapKV(Li et al., 2024a) discard tokens based on specific heuristics. However, once discarded, these tokens are no longer available for future generations. This limitation is particularly problematic in scenarios such as multi-turn chat or multiple QA sessions on a fixed document, where it is essential to access different parts of the document in each turn or question.

**Estimating top-k attention scores via partial computation** Attention scores are one of the critical components that determine the contribution of a token to the output of the attention mechanism. Identifying the top-k tokens with the highest attention scores requires $O(nd)$ computation, as it involves the dot product of the query with each token in the KV cache, where $n$ is the number of tokens in the KV cache and $d$ is the dimensionality.

Double Sparsity (Yang et al., 2024) reduces this computational cost by selecting fewer channels to estimate dot products, which are then used to identify the top-k tokens. The channels are chosen based on offline calibration of channel norms. Loki (Singhania et al., 2024) reduces the dimension by using traditional dimensionality reduction techniques to obtain low-dimensional representation for keys. InfLLM (Xiao et al., 2024) and Quest (Tang et al., 2024) reduce computation along the $n$-dimension by creating page-level representations. These methods include or exclude all tokens on a page at once. While these approaches are effective, they often fail to provide high recall for the top-k tokens with respect to attention scores.

**Retrieval Algorithms for Top-k** Recently, RetrievalAttention (Liu et al., 2024a) proposed using a graph-based nearest neighbor algorithm to identify the top-k tokens with the maximum inner product. RetrievalAttention offloads the top-k computation to the CPU due to the sparse computations involved with graphs, leading to additional latency. SqueezedAttention (Hooper et al., 2024), a concurrent work with ours, proposed to solve the top-k problem efficiently by clustering keys. In contrast, we use succinct bit signatures

from key-value and queries to identify pivotal tokens.

**Estimating attention output using Hash tables** Learning-to-hash attention (Sun et al., 2021) proposes to learn hash table mappings to obtain efficient attention with sparse retrieval. They design a learning algorithm to obtain a balanced distribution of keys in the hash table buckets. MagicPig(Chen et al., 2024) noted that top-k attention is biased and can be problematic in cases when attention scores are relatively balanced. They propose importance sampling-based unbiased attention estimation. They use LSH tables to approximate the importance-sampling distribution. These methods can be potentially combined with HashAttention for further efficiency.

## 3. Background

### 3.1. Scaled Dot Product Attention (SDPA)

The computation of SDPA for key and value embeddings, $\mathbf{K}, \mathbf{V} : n_1 \times d$, and $\mathbf{Q} : n_2 \times d$ can be written as,

$$\text{SDPA}(\mathbf{K}, \mathbf{V}, \mathbf{Q}) = \text{softmax}\left(\frac{\mathbf{Q}\mathbf{K}^\top}{\sqrt{d}}\right)\mathbf{V}$$

If $n_2 = 1$, then it can be simplified into,

$$\text{SDPA}(\mathbf{K}, \mathbf{V}, \mathbf{q}) = \sum_{i=1}^{n_1} (a_i \mathbf{V}[i])$$

where $a_i = \frac{\exp\langle \mathbf{K}[i], q\rangle}{\sum_{j=1}^{n} \exp\langle \mathbf{K}[j], q\rangle}$ are called attention scores.

### 3.2. Learning Recommendation Model

The recommendation problem can be abstracted as: Given a set of items $\mathcal{I}$ and a user $u \in \mathcal{U}$, the recommendation aims to select a small subset of $\mathcal{I}$ that is relevant to $u$. The historical relevance of items for users is captured in an interaction matrix with shape $|\mathcal{U}| \times |\mathcal{I}|$. Traditionally, the learning of recommendation model has been cast as a matrix factorization of the interaction matrix. The two matrices, thus obtained, provide us with embeddings of items and users that can be used for inference. Subsequently, auxiliary information such as description, profile, interaction history, etc., and deep learning models were used to obtain richer user and item representations. Alternatively, one can learn the embeddings so that the embeddings of the relevant items lie close to the user embeddings in terms of $l_p$ distance. To improve the efficiency of relevant item retrieval, approximate near-neighbor algorithms and data structures are utilized on top of the embedding space. Sparsifying attention is a recommendation problem with key-value pairs as items and queries as a user. We want to select key-value pairs that minimize the error of output embedding w.r.t full attention for a particular query.

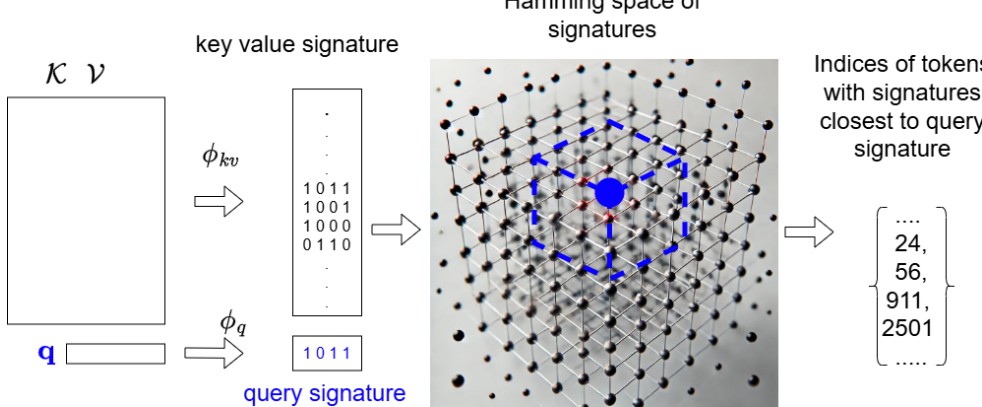

*Figure 2.* HashAttention working: The key-value pairs are mapped to bit signatures via learned mapping function $\phi_{kv}$. The query is mapped to bit signature via function $\phi_q$. The tokens closest to the query signature are chosen as candidates for attention computation.

## 4. HashAttention

The general recipe of sparse attention can be viewed as a combination of three subroutines (1) SCORE (2) TOPK and (3) GATHER-ATT .

SCORE $(\mathcal{K}, \mathcal{V}, \mathbf{q})$: Given a query $\mathbf{q}$ and a set of tokens $\mathcal{K}, \mathcal{V}$, SCORE assigns a score to each token in $(\mathcal{K}, \mathcal{V})$. TOPK is the standard top-k on these scores. GATHER-ATT $(\mathcal{K}, \mathcal{V}, \mathbf{q}, \mathcal{I})$: Given a set of tokens $\mathcal{K}, \mathcal{V}$, a query $\mathbf{q}$ and set of indices $\mathcal{I}$ identifying pivotal tokens, GATHER-ATT computes the attention only using tokens indicated by $\mathcal{I}$ as follows,

$$\text{GATHER-ATT } (\mathcal{K}, \mathcal{V}, \mathbf{q}, \mathcal{I})$$
$$= \sum_{i=1}^{|\mathcal{K}|} \frac{\mathbf{1}(i \in \mathcal{I}) \exp\left(\langle \mathbf{q}, \mathbf{k}_i \rangle\right) \mathbf{v}_i}{\sum_{j=1}^{|\mathcal{K}|} \mathbf{1}(i \in \mathcal{I}) \exp\left(\langle \mathbf{q}, \mathbf{k}_j \rangle\right)}$$

where $\mathbf{v}_i$, $\mathbf{k}_i$ refer to $i^{th}$ embeddings of $\mathcal{V}$ and $\mathcal{K}$ respectively, and $\mathbf{1}$ is the indicator function. In the text, we will overload the function SCORE $(\mathbf{k}, \mathbf{v}, \mathbf{q})$ to denote the score assigned to a single key for methods that work on each key individually.

Most sparse-attention methods can be viewed as a combination of the above three routines with differences in the SCORE function. Following the SCORE routine, sparse-attention methods can be implemented using identical routines for TOPK and GATHER-ATT . Furthermore, we find in experiments, that most time is spent in GATHER-ATT and TOPK routines. Thus we should devise a good quality SCORE function with reasonably small latency. The quality of SCORE can be measured in terms of its recall$(n, k)$ defined as follows. Let $\mathcal{I}_n$ be the set of top $n$ indices chosen by SCORE function and $\mathcal{I}_{true,k}$ be the set of $k$ true pivotal tokens. Then, the recall is computed as

$$\text{recall}(\mathcal{I}_n, \mathcal{I}_{true,k}) = \frac{|\mathcal{I}_n \cap \mathcal{I}_{true,k}|}{|\mathcal{I}_{true,k}|}$$

The SCORE function has three important aspects.

1. **Auxiliary memory**: Various methods such as Double Sparsity, Quest, and InfLLM including HashAttention use meta representations of keys/tokens that are cached for efficient score computation. Since memory is an important bottleneck with huge sizes of KVCache, the amount of memory used for meta-information is an important consideration.
2. **Latency** is another important factor. Since SCORE is generally not a latency bottleneck, one can devote more computation to obtain higher-quality retrieved tokens
3. **Quality**: A SCORE with higher recall, implies we can effectively use fewer tokens in GATHER-ATT which will further improve the overall efficiency of the attention procedure.

### 4.1. The SCORE function for HashAttention

Identifying the top tokens for a query is analogous to a user-item recommendation problem. HashAttention uses two learnable mapping functions, $\phi_{kv} : R^{2d} \rightarrow [0, 1]^b$, $\phi_q : R^d \rightarrow [0, 1]^b$ to lift key-value and queries to hamming space of dimension $b$. In this space, we can efficiently compare a given query to the keys using hamming distance $\mathcal{H}$. The SCORE function is defined by

$$\text{SCORE } (\mathbf{k}, \mathbf{v}, \mathbf{q}) = -\mathcal{H}(\phi_{kv}(\mathbf{k}, \mathbf{v}), \phi_q(\mathbf{q})) \quad (1)$$

For $\phi_{kv}$ and $\phi_q$, we use independent feed-forward network $\mathcal{F}$, followed by a sign function to extract bits. The general function $\phi$ can be written as,

$$\phi(\mathbf{x}) = \text{relu}(\text{sign}(\mathcal{F}(\mathbf{x}))) \quad (2)$$

The bits are packed into an integer. Let $\phi_{\text{int}}$ be the compete mapping function.

**Implementation:** The integer signatures for key-values are computed and cached along with KV Cache. During

the decoder hot path, we compute the query signature and perform bit operations to compute the hamming distance between the query and key-values.

$$\mathcal{H}(\phi_{kv}(\mathbf{k}, \mathbf{v}), \phi_q(\mathbf{q}))$$
$$= \text{bitcount}(\text{bitwise\_xor}(\phi_{\text{int},kv}(\mathbf{k}, \mathbf{v}), \phi_{\text{int},q}(\mathbf{q})))$$

In our experiments, we focus on $\phi_{kv}$ that acts only on the key vector $\mathbf{k}$ and ignore the $\mathbf{v}$ vector or its features such as norm. Incorporating $\mathbf{v}$ in $\phi_{kv}$ is left for future exploration.

## 4.2. HashAttention for adapting to pre-trained LLMs

To use HashAttention for existing SDPA attention in pre-trained models, SCORE function of HashAttention should align with the best selection of tokens w.r.t the final attention computation. The best ordering for an SDPA attention is presented in the lemma below.

**Lemma 4.1.** *Consider $n$ KV embeddings $\mathcal{K}, \mathcal{V}$ and query $\mathbf{q}$. Let $\mathbf{a}$ be the attention scores of tokens w.r.t query $q$. Then the contribution of a token $i$, assuming no other token is used, towards the final output is proportional to*

$$a_i||\mathbf{v}_i||_2 \qquad (3)$$

The proof is presented in appendix B.1. We can use top tokens identified by this score to train our HashAttention modules for each attention. We pose HashAttention training as a classification problem where each embeddings generated in HashAttention are used to predict the top-$k$ tokens of the attention head that it is associated with. We choose $k$ to be a small number such as $64$. We use binary cross-entropy loss in a multi-class setting to train our functions $\phi_{kv}$ and $\phi_q$ with a standard Adam optimizer. We mention some important details below,

**Class imbalance** As the context length increases, the class imbalance for classification increases. For instance at 64,000 context length, while using top-64 tokens to predict, only 0.1% of labels are of class 1. We use class weights to resolve the issue of class imbalance. Since the imablance depends on the context length. We use the following formula to compute the class 1 weights, parameterized with $\alpha$ and $\beta$:

$$\text{class1-weight} = \alpha + \beta\text{context-length} \qquad (4)$$

$\alpha$ and $\beta$ are hyperparameters that can be chosen.

**Soft-partitioning** While we use the sign function to perform space-partitioning and obtain bits in $\phi$ mapping computation during inference, we use the tanh function in place of the sign function as a softer version of partitioning while training.

**Training dataset** We find that HashAttention trained on a generic dataset works well on a variety of tasks. To further improve the quality, we can finetune HashAttention on task-specific data. Additionally, we find that HashAttention trained on shorter sequences (say $<= 64K$) do not naturally scale to longer sequences (e.g. $128K$). So, HashAttention needs to be trained for appropriate context-lengths.

We run the LLM model in inference mode in a chunk-based fashion, i.e. the tokens are processed in chunks. At the end of each inference, all HashAttention modules are independently and locally trained for one step on queries from this chunk and previously cached KV embeddings.

## 4.3. Theoretical underpinnings of HashAttention

In this section, we briefly explain why HashAttention works. Lemma 4.1 shows that analyzing token importance in isolation, the contribution of $i^{th}$ token is proportional to $a_i||\mathbf{v}_i||_2$ where $a$ and $\mathbf{v}$ are attention score and value vector respectively. Identifying tokens with this true score can be cast as a maximum inner product search problem as shown below,

**Lemma 4.2.** *Consider a KV Cache $\mathcal{K}, \mathcal{V}$, and query vector $\mathbf{q}$. The ordering imposed on scores $a_i||\mathbf{v}_i||_2$ for $i^{th}$ token is same as the ordering imposed by inner product $\langle [\mathbf{q}, 1], [\mathbf{k}, log(||\mathbf{v}||_2)] \rangle$ where [] denotes concatenation.*

The proof is presented in appendix B.1. The Maximum inner product search problem can be reduced to a cosine-similarity search problem (Shrivastava & Li, 2014; Neyshabur & Srebro, 2015) using asymmetric transformation on query and key-values. Combining Lemma 4.1 and Lemma 4.2 along with results from (Neyshabur & Srebro, 2015), we can equivalently determine ordering based on cosine similarity between the following vectors,

$$\psi_{kv}(\mathbf{k}, \mathbf{v}) = \left[\mathbf{k}, \log(||\mathbf{v}||_2), \sqrt{M^2 - ||[\mathbf{k}, log(||\mathbf{v}||_2)]||^2}\right]$$

where M is the maximum norm of all vectors $[\mathbf{k}, \log(||\mathbf{v}||)]$

$$\psi_q(\mathbf{q}) = [\mathbf{q}, 1, 1]$$

Cosine similarity can be approximated using bit signatures obtained by random signed projections (Gionis et al., 1999). HashAttention is motivated by this theoretical understanding of similarity search. Furthermore, we use learnable functions in place of $\psi$ to exploit the patterns in query and key distribution and obtain succinct bit signatures. Using independent encoding functions for key-values and queries helps in solving the MIPS problem.

## 4.4. Efficiency of token level sparsity.

As opposed to methods such as Quest or InfLLM that use block sparsity in attention, HashAttention uses token-level sparsity. We want to note that token-level sparsity is unavoidable when trying to implement sparse attention during inference time when training assumes full attention, because

*Table 1.* A head-to-head comparison of sparse-attention baselines on datasets from LongBench from all the categories using the LLama-3.1-8B-Instruct. InfLLM, DS and Quest use auxiliary memory for shorter representation of keys – noted in Aux-budget as bits per token. HashAttention while having smallest auxiliary budget outperforms all baselines on average. HashAttention is trained with mixture of openwebtext dataset and LongBench datasets (25 examples per dataset). Evaluation is done on first 175 examples (MFQA was excluded from training since it has only 150 samples)

| | Category → | | MQA | SQA | Summ | FS-Learn | Synthetic | Code | |
|---|---|---|---|---|---|---|---|---|---|
| Model | Aux:bits/token | Tokens | HPQA | MFQA | QmSm | TQA | PassR | RepoB | Average |
| Full Model | NA | NA | 54.83 | 55.17 | 24.91 | 91.31 | 100.00 | 55.07 | 63.55 |
| Oracle(top) | NA | 512 | 52.10 | 53.45 | 25.14 | 91.39 | 100.00 | 58.49 | 63.43 |
| H2O | NA | 512 | 36.62 | 26.61 | 17.85 | 80.75 | 43.43 | 55.55 | 43.47 |
| StreamLLM | NA | 512 | 33.32 | 27.98 | 17.93 | 51.95 | 11.43 | 57.07 | 33.28 |
| InfLLM | 256(pg=32,bit=16) | 512 | 47.75 | 51.99 | 23.14 | 88.32 | 33.00 | 42.99 | 47.87 |
| InfLLM | 256(pg=16,bit=16) | 512 | 48.27 | 53.09 | 22.90 | 88.88 | 32.81 | 43.45 | 48.23 |
| DS | 32(ch=16,bit=2) | 512 | 50.39 | 50.57 | 23.41 | 90.32 | 98.86 | 57.72 | 61.88 |
| DS | 64(ch=16,bit=4) | 512 | 52.60 | 53.44 | 23.75 | 90.44 | 99.43 | 57.01 | 62.78 |
| DS | 128(ch=8,bit=16) | 512 | 41.50 | 43.21 | 21.80 | 86.83 | 82.29 | 52.80 | 54.74 |
| Quest | 32(pg=16,bit=2) | 512 | 53.13 | 51.31 | 23.01 | 90.15 | 98.29 | 58.29 | 62.36 |
| Quest | 64(pg=16,bit=4) | 512 | 52.51 | 54.35 | 23.75 | 91.50 | 98.86 | 59.03 | 63.33 |
| Quest | 128(pg=32,bit=16) | 512 | 51.60 | **53.60** | 22.94 | 90.53 | 97.71 | 57.06 | 62.24 |
| HashAttention | 32 | 512 | **54.08** | 53.35 | **25.08** | **92.41** | **100.00** | **59.98** | **64.15** |

*Table 2.* Llama3.1-8B-Instruct with HashAttention-16× on LongBench . $AVG^{\bar{pc}}$ is average without passage count

| | multifieldqa-en | passage-retrievel-en | samsum | qasper | musique | gov-report | multi-news | trec | hotpotqa | passage-count | qmsum | 2wiki | triviaqa | narrativeqa | lcc | repobench-p | AVG | $AVG^{\bar{pc}}$ |
|---|---|---|---|---|---|---|---|---|---|---|---|---|---|---|---|---|---|---|
| Full | 54.68 | 100 | 43.75 | 45.80 | 29.41 | 34.90 | 27.11 | 71.43 | 55.40 | 8.29 | 25.14 | 45.62 | 90.55 | 30.19 | 62.87 | 55.31 | 48.78 | 51.48 |
| 16x | 53.65 | 100 | 42.82 | 43.52 | 29.16 | 34.99 | 26.68 | 70.29 | 51.92 | 1.85 | 24.75 | 42.97 | 92.14 | 27.93 | 64.89 | 60.49 | 48.00 | 51.08 |

*Table 3.* Llama3.1-8B-Instruct with HashAttention-16× on RULER@16K

| | niah_single_1 | niah_single_2 | niah_single_3 | niah_multikey_1 | niah_multikey_2 | niah_multikey_3 | niah_multivalue | niah_multiquery | vt | cwe | fwe | qa_1 | qa_2 | Average |
|---|---|---|---|---|---|---|---|---|---|---|---|---|---|---|
| Full | 100 | 100 | 100 | 100 | 100 | 100 | 99.25 | 98.75 | 99.2 | 50.5 | 90 | 87 | 54 | 90.66 |
| 16 × | 100 | 100 | 100 | 99 | 90 | 96 | 98.5 | 98.75 | 93 | 62.4 | 87.33 | 82 | 57 | 89.53 |

important tokens, defined by higher attention scores, may not appear in the same block, and not using some important tokens to adhere to an imposed block size will only lead to an inferior top-k approximation. Interestingly, and contrary to popular belief, token-level sparsity does not lead to system inefficiency.

HashAttention implementation is built upon the vLLM page attention framework (Kwon et al., 2023), where each token corresponds exactly to one page (i.e., page size = 1). After identifying top-k tokens, we use their indices without physically moving or gathering these tokens into a separate memory buffer. The page attention kernel explicitly utilizes these indices to selectively compute attention only for the specified tokens, efficiently ignoring irrelevant tokens. The page attention kernel is highly optimized for GPU memory access patterns. Due to the GPU cache line size of 128 bytes, optimal memory bandwidth utilization is achieved as long

as contiguous data access meets or exceeds this cacheline size. It is important to note that many state-of-the art inference framework implement attention using paged-attention backbone with page size 1. They find that with correct optimization, the efficiency does not depend on page size (Ye et al., 2024).

## 5. Experiments

In the experiments below, we use two versions of HashAttention. HashAttention is trained using openwebtext(Gokaslan et al., 2019) dataset with samples concatenated together to obtain the required context length. HashAttention* is further finetuned from HashAttention for specific dataset.

**baselines:** StreamingLLM (Xiao et al., 2023), H2O (Zhang et al., 2023), InfLLM (Xiao et al., 2024), DoubleSparsity (DS) (Yang et al., 2024), and Quest (Tang et al., 2024).

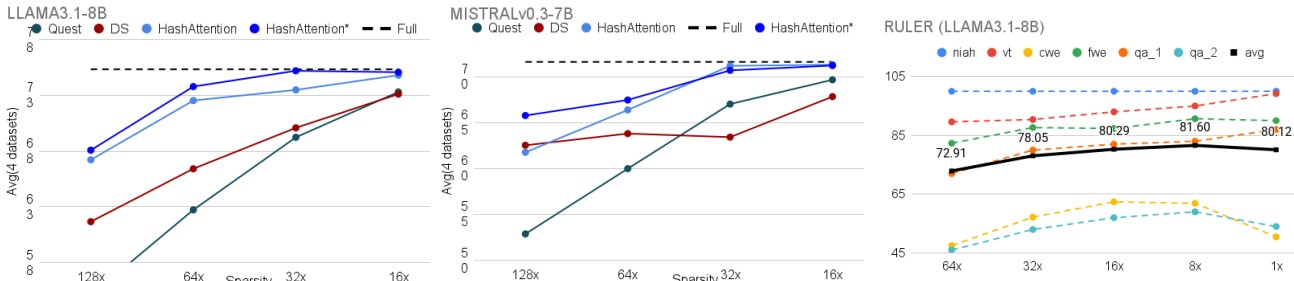

*Figure 3.* Pareto Curves for **[LongBench]** Average quality of passage_retreival_en, triviaqa, multifieldqa_en, and hotpotqa from Longbench at different sparsities at auxiliary budget of 32bits PTPA. HashAttention almost reaches the full model quality at $16\times$ for Llama-3.1-8B-Instruct and $32\times$ for Mistral-7B-v0.3-Instruct. HashAttention* further improves the Pareto curves to obtain full model quality at $32\times$ for LLAMA-3.1-8B-Instruct. Detailed dataset results are deferred to Figure 6 in Appendix. **[RULER@16K]** Quality of Llama-3.1-8B-Instruct on RULER@16K benchmark at varying sparsity levels. On average, HashAttention-$16\times$ achieves the same quality as the full model. Interestingly for most datasets, sparsity at some level helps improve over the full model (4/6 datasets considered show this at sparsity $>= 8\times$)

We also include Oracle which predicts exact top-k. For all methods, we include the first and most recent 128 tokens to ensure that the important parts of the prompt are always in context. In cases where methods use auxiliary memory (e.g. label cache in DS and bit signatures in HashAttention), we denote it as bits per token per head(PTPA). The token budget refers to the number of heavy tokens retrieved. InfLLM and Quest are page-based retrievers. These baselines have two parameters: page size (`pg`) and the number of representative vectors (`rep`). In InfLLM, we always set `rep=4`. While these baselines do not inherently incorporate further quantization, we apply additional quantization to reduce auxiliary memory usage, as is done with DoubleSparsity (DS). DS has two parameters: the number of channels (`ch`) and the number of quantization bits (`bit`). Baselines are further explained in Appendix A.

**prompt-offset:** To simulate a scenario of preprocessed long context with relatively smaller prompts, without relying on specific datasets, we apply sparse attention only to the last `offset` tokens of the complete prompt and all subsequent generations. We use `offset` of 128 for Longbench and 32 in RULER.

**Auxiliary Memory:** Since memory is a significant bottleneck in deploying LLM models, we aim to evaluate different methods under the same auxiliary memory budget of 32 bits PTPA unless stated otherwise.

**Models and datasets:** We use Llama-3.18B-Instruct(Dubey et al., 2024) and Mistral-7B-Instruct-v0.3(Jiang et al., 2023) models and Longbench(Bai et al., 2024b) and RULER(Hsieh et al., 2024) datasets in our evaluation. All attention heads are replaced with sparse attention.

**Efficiency frameworks:** We use GPT-FAST(GPTFast, 2024) and FlashDecode(Dao, 2023) to test improvements

offered by HashAttention.

### 5.1. Quality with HashAttention

We present the following experiments to evaluate the quality of HashAttention.

**A head-to-head comparison of HashAttention at fixed budget** For this experiment, we randomly select one dataset from each category in the LongBench and use the first 175 samples (the last 25 samples are added to the training set). We set a budget of 512 pivotal tokens for all baselines. The results are presented in the Table 1.

**Pareto curves** To get a broader comparative picture, we use four datasets (Passage Retrieval (English), TriviaQA, HotpotQA, and MultifieldQA (English) ) from Longbench and six datasets (niah, vt, cwe, fwe, qa1, and qa2) from RULER (16K) to perform a more elaborate Pareto analysis. We compare against Quest and DS at 32 PTPA auxiliary budget. Unless explicitly mentioned, we use HashAttention (and not HashAttention* ). These results are presented in Figure 3.

**Full benchmark at $16\times$ sparsity** We also present HashAttention at $16\times$ sparsity on full LongBench and RULER benchmarks for Llama-3.1-8B-Instruct. The results are presented in Table 2 and Table 3.

**Micro benchmarking the recall rates** To show that the improvements in quality come from the improvement in pivotal token retrieval, we show the recall rates of top-32 tokens at different context lengths for the Llama-3.1-8B-Instruct model. The result is presented in Figure 5 in the appendix.

We make the following observations,

- Table 1 presents the landscape of different sparse-

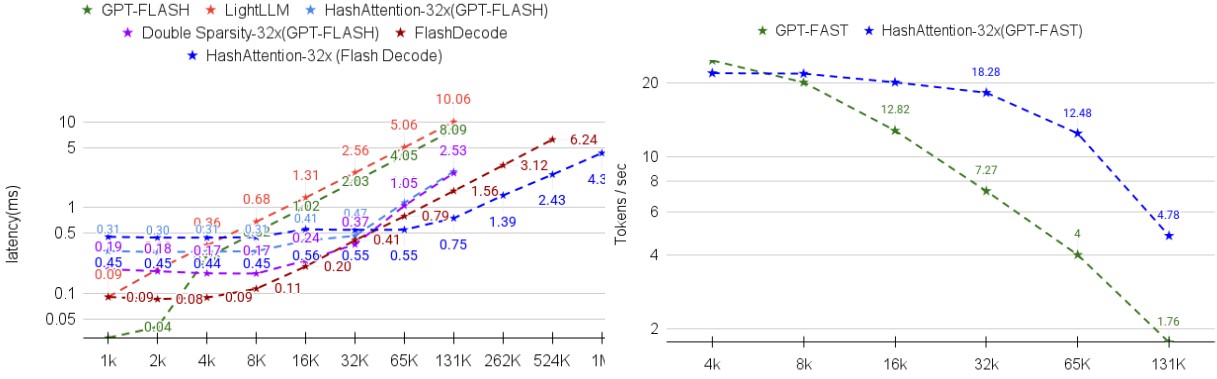

(a) Latency of various kernels at different context lengths

(b) Throughput (batch=1) in GPT-FAST inference system at different context lengths

*Figure 4.* Efficiency improvements by incorporating HashAttention in GPT-FAST and FlashDecode. HashAttention-32x shows improvement in latency and end-to-end throughput on sequences longer than 8K in GPT-FAST and 65K in FlashDecode

attentions. H2O and StreamingLLM fail due to the incorrect fixed sparsity assumption. InfLLM uses dynamic sparsity, but the heuristic-based scoring does not generalize well over all tasks leading to poor performance. Quest and DS perform relatively well on all tasks. However, it needs more auxiliary memory to obtain good results. Overall, HashAttention outperforms all the baselines while using minimum auxiliary memory.

- Figure 3, Table 2, and Table 3 show that HashAttention can significantly reduce the number of tokens in attention computation. HashAttention is almost $4\times$ more token-efficient compared to Quest and DS at an auxiliary budget of 32 bits/token. Overall, HashAttention seems to preserve most of the model quality at $16\times$ sparsity (0.78 points less on Longbench, 0.4 points without the worst performing dataset, and 1.13 points less on RULER). The results can be further improved using HashAttention*.

- Certain levels of sparsity seem to improve quality over the full model in certain tasks. This could be potentially explained as sparse attention avoids attention to distractor tokens.

- HashAttention trained on English data also performs reasonably well on Chinese datasets in LongBench, although with a higher drop in quality. (2.66 points on average. See table 5 in the appendix.) The results on coding tasks (lcc and repobench-p in Longbench) are especially better than the full model. Coding tasks might be more sensitive to distractors than general English tasks.

- The recall of pivotal tokens shown in Figure 5 (in appendix) gives an insight into why HashAttention outperforms other baselines. The reason is that HashAttention is better at retrieving pivotal tokens.

### 5.2. Efficiency with HashAttention

To evaluate the efficiency that HashAttention has to offer, we perform the following experiments. In these experiments, we use 3-layered MLP (128x128-128x128-128x32) for mappings learned in HashAttention. Using smaller MLPs will improve latency results for HashAttention. We use $32\times$ sparsity for these experiments.

- We incorporate HashAttention in the GPT-FAST attention kernel and FlashDecode attention kernel and measure the latency improvements.

- We incorporate HashAttention in the GPT-FAST inference system to measure the end-to-end improvement in throughput.

We make the following observations.

- The matrix multiplication in mapping functions used to obtain bit-signatures dominates the latency up to 8K context length in GPT-FAST and up to 65K in Flash Decode. Since the main difference between the two attention implementations is that FlashDecode uses sequence parallelism, whereas GPT-FAST does not, this behavior is expected. Post 4K context length for GPT-FAST and 65K context length for FlashDecode, HashAttention improves the latency for the attention. We can see up to $4.3\times$ improvement in GPT-FAST and $2.54\times$ improvement in FlashDecode.

- The throughput improvements can be seen starting at 8K context length. The throughput improvement ( ratio of the throughputs) starts declining as sequence length increases. This can be attributed to TOPK operation becoming expensive at higher sequence lengths. In this experiment, we see the best throughput gains of $3.12\times$ occur at a sequence length of 32K for $32\times$ sparsity.

## 5.3. Ablations

In this section, we present ablations. The tables / figures for this section are presented in Appendix.

**HashAttention Bit Signatures vs. LSH Bit Signatures:**
We compare LSH bit signatures (Gionis et al., 1999), obtained via random signed projections, with learned bit signatures used in HashAttention. We find that LSH requires significantly more bits to perform reasonably well on top-k prediction. This inefficiency stems from the agnostic nature of LSH. The results are presented in Table 8. While HashAttention performs well with 32-bit signatures, LSH fails to achieve comparable performance even with more than 1000 bits.

Note that this is not a direct comparison to MagicPig (Chen et al., 2024), as MagicPig builds LSH tables from bit signatures for sparse retrieval and modifies the attention computation to incorporate the sampling view of LSH. Nevertheless, it provides a reasonable indication of the auxiliary memory used by MagicPig to achieve sparse attention.

**Query and Key distributions under HashAttention embedding** While HashAttention does not directly aim to resolve the out-of-distribution issue between queries and keys, the learned mappings bring the distributions closer due to the nature of the training loss. We measure the empirical cosine similarity between each query and its top-32 tokens using the original embeddings, HashAttention (signed) embeddings, and HashAttention (tanh) embeddings used during training. We find that the cosine similarity between key and query embeddings is significantly higher when using HashAttention embeddings compared to the original embeddings. The results are presented in Table 7.

**Quality vs. bit-width in HashAttention** As expected, we find that using more bits helps with the quality of top-k prediction. The cross-entropy loss of top-32 prediction using different bit-width is presented in Table 9.

**Latency of SCORE computation vs. full inner product.**
Since increasing bit-widths can improve attention quality, it is reasonable to ask how many bits can be used before the computational advantage of HashAttention disappears. We present this ablation in Table 6. We find that even for bit-widths as large as 512, the latency of SCORE computation using our optimized kernel—which employs bitwise operations—is still lower than that of full inner product computation for 128-dimensional vectors with float16 precision.

## 6. Limitations and Discussion

We discuss the limitations of the proposal and evaluation here. HashAttention needs to be trained at least once. As opposed to training-free methods this is a limitation since every model needs its own HashAttention. This is, however, not very different from the calibration used in DS. To obtain better results, we can use further finetuning on the task-specific data. Practically, this is okay for users who care about a specific application of LLM. In terms of evaluation, we restrict our evaluation to situations where KV Cache is on the GPU. For extremely long context situations, KV Cache needs to be stored on the CPU RAM, and evaluation of HashAttention in such a scenario using flash-decode would be most valuable. However, such an evaluation is out of the scope of this paper. Having said that, HashAttention clearly shows superior quality, larger sparsity, and lower auxiliary memory compared to different top-k-based sparse attentions under identical conditions.

## 7. Conclusion

HashAttention proposes a principled approach to sparsify attention based on hashing key-value pairs and queries to a compressed semantic space. Near neighbor comparisons can be efficiently performed in this semantic space using bit operations. The quality of tokens retrieved by HashAttention, under similar resources, is significantly superior to other sparsity baselines, leading to upto $4\times$ further reduction of tokens as compared to leading sparse attention baselines. Overall, we see reduction in KV Cache usage upto $16-32\times$ with minimal drop in quality (within 1 point) across benchmarks and models, leading to latency and throughput gains.

## Acknowledgments

Sky Computing Lab is supported by gifts from Accenture, AMD, Anyscale, Cisco, Google, IBM, Intel, Intesa Sanpaolo, Lambda, Lightspeed, Mibura, Microsoft, NVIDIA, Samsung SDS, and SAP. Aditya Desai's position at sky-lab is supported by Mayfield Charitable.

## Impact Statement

This paper presents work whose goal is to advance the efficiency of Machine Learning which can lead to wider adoption of large language models. There are many potential societal consequences of advancing machine learning and its efficiency, none of which we feel must be specifically highlighted here.

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

# A. Baselines

The primary purpose of HashAttention is to sparsify attention to improve the memory loading and computation of attention computation. Thus we consider approaches in literature with similar goals.

## A.1. StreamingLLM

This attention was developed for processing streams of text. There are two different challenges when it comes to ingesting stream of text with existing LLM models. Firstly, most LLM models are trained on restricted context length and thus do not support longer texts. Secondly, the computational cost of decoding grows linearly with context length leading to prohibitive computation. StreamingLLM proposes to keep two portions of the context – the first (also called attention sink and the local context. Thus it sparsifies the attention computation.

Since StreamingLLM categorically ignores a major chunk of context. It does not perform well on long context benchmarks which needs the model to consider all the context.

Hyperparameters: sink size and local size.

## A.2. ScissorHands / H2O

This attention was developed primarily to reduce the memory storage of KV Cache – goal very well aligned with sparsification of attention and reducing memory loading during decoding. However, the setting used in ScissorHands / H2O is more restrictive since, the decisions are made in a streaming fashion and tokens evicted can never be retrieved unless recomputed. The idea in ScissorHands / H2O is that if some tokens in context have not been useful in recent predictions then they are not going to be useful in the future generations and can be dropped. In our experiments we use H2O since they have easier codebase.

Scissorhands and H2O both heuristically drop the tokens. The tokens dropped at one point are not available subsequently. This is clearly an issue in different settings such multi-turn chat etc. It should be noted that the proposal of ScissorHands and H2O are for reducing decoding time monologue of LLM. In that particular setting the proposals are useful. But their effectiveness is also restricted to that setting.

Hyperparameters: sink size, local size and token budget.

## A.3. Retrieval Attention

In Retrieval Attention, the attention computation is preceded by top-k computation using approximate near neighbor search algorithms and full attention is computed on estimated top-k. Most graph based algorithms (including the one proposed in Retrieval Attention) need to be run on CPUs due to their irregular computation pattern. Thus, Retrieval Attention by default always stores the KVCache on CPU.

This is a close cousin of HashAttention. The motivation of both methods is identical in the sense that attention can be replaced by approximate near neighbour search. RetrievalAttention uses traditional graph based search algorithm to find near nieghbours, whereas HashAttention uses learning to hash to create a quantized similarity space for retrieval. A major drawback of RetrievalAttention is that it is not GPU friendly which causes indexing and querying to be slower for large contexts.

Hyperparameters: sink size, local size and ROAR graph hyper parameters.

## A.4. InfLLM

InfLLM maintains the attention sink and local context as with streaming LLM. Additionally, it also maintains the tokens in between and retrieve chunks from them. It divides the KVCache into contiguous chunks and from each chunk a few representative keys. These keys are used to compute the importance of the chunks w.r.t a given query. Top few chunks are chosen for final attention computation via full computation. In order to choose top scoring chunks, the paper proposes performing full dot product computation can be performed with representative vectors which can also be replaced by off-the-shelf near neighbour computation on CPUs.

This again is similar to the setup of HashAttention. The chunk based treatment for retrieval reduces the computational cost

of computing the relevant chunks. However, the heuristic way of computing the representative keys can lead to issues of missing key chunks while retrieving. Apart from that, the method is identical to RetrievalAttention.

Hyperparameters: sink size, local size, token budget, page size, number of representative vectors

### A.5. Quest

Quest is similar to InfLLM with the difference being the computation of the importance of a chunk. Quest maintains a $\max$ and $\min$ vectors for each chunk which maintains co-ordinate wise extremes of the chunk. These are used to estimate the maximum possible inner product within a chunk given a query.

It is clear to see that if we restrict the token budget, we might end up retrieving falsely important chunks at the cost of discarding important ones. We see this in our experiments as well.

Hyperparameters: sink size, local size, token budget, page size

### A.6. Double Sparsity

Double sparsity chooses 16 coordinates from the 128 dimensions of key embeddings via offline calibration and use them for computing top-k. Then full attention is computed on these top-k. This 16 dimensional slice of K Cache is called as label cache and can be further quantized to 4 bits without much impact on accuracy.

This again is similar to the setup of HashAttention and RetrievalAttention – the difference being how to compute the top-k. Surprisingly the 16 channels (16x16 = 256 bits) identified by Double Sparsity are good top k indicators. In comparison HashAttention uses 32 bit signatures and uses bit operations for manipulating signatures.

Hyperparameters: sink size, local size, token budget, label size.

***Quality and computation of sparsity with HashAttention vs. baselines***

The choice of completely ignoring parts of context in streamingLLM, heuristic based permanent eviction of Scissorhands/H2O, and heuristic based representative key selection of InfLLM causes these approaches to be inferior to HashAttention. Retrieval Attention, Double Sparsity and HashAttention all are based on determining the top-k and using it for attention computation. Thus, the quality depends on ANN algorithm used. In terms of computational complexity, HashAttention and Double sparsity can be run on GPU and thus are faster for reasonably sized context lengths as compared to Retrieval Attention. Additionally, HashAttention only uses an integer signature for computation of top-k which is memory and compute effective as compared to Double sparsity

## B. Lemma Proofs

### B.1. Lemma 1: Best Sparse solution for adaptation for 1-token selection

Using the following notations, $\mathbf{V} : n \times d$ value matrix, $\mathbf{a} : n \times 1$ attention scores. $\mathbf{S} : n \times n$ diagonal sampling matrix **with exactly one non-zero** in the diagonal (selection of a single token). Note that this ignores the cross token corrleation in value vectors. Then under the sampling the final embedding is,

$$a^\top S V \tag{5}$$

The residual is $||a^\top V - a^\top S V||_2$. Then the best sampling is the one that minimizes the

$$S = \operatorname{argmin}||a^\top V - a^\top S V||_2^2 \tag{6}$$

$$||a^\top V - a^\top S V||_2^2 \tag{7}$$
$$= (a^\top(I - S)V)(a^\top(I - S)V)^\top \tag{8}$$
$$= a^\top(I - S)VV^\top(I - S)a \tag{9}$$
$$= \sum_{i=1}^{n}(1 - \mathbf{1}_{Si})a_i^2||V_{i,:}||^2 \tag{10}$$

To minimize the residual, we have to choose, $\mathbf{1}_{Si}$ to be 1 which have higher $a_i||V_i||$

## B.2. Lemma 2: Identifying topk w.r.t isolated token score is a MIPS problem

$$a_i||v_i||_2 = \frac{e^{\langle \mathbf{q},\mathbf{k}\rangle}||v_i||_2}{\sum e^{\langle \mathbf{q},\mathbf{k}\rangle}} \tag{11}$$

$$= \frac{e^{\langle \mathbf{q},\mathbf{k}\rangle+\log(||\mathbf{v}_i||)}}{\sum e^{\langle \mathbf{q},\mathbf{k}\rangle}} \tag{12}$$

$$\propto \langle \mathbf{q},\mathbf{k}\rangle + \log(||\mathbf{v}_i||) \tag{13}$$

$$= \langle [\mathbf{q},1],[\mathbf{k},\log(||\mathbf{v}_i||)]\rangle \tag{14}$$

## C. Other ablations

### C.1. Recall rates of HashAttention and DS

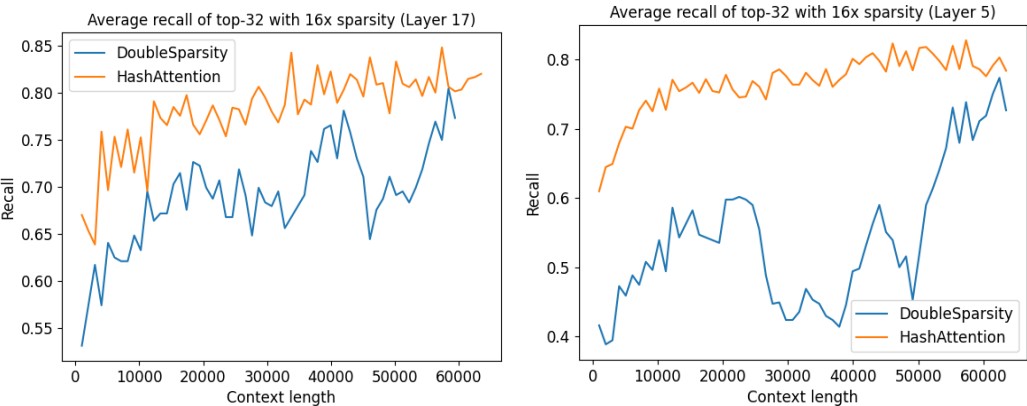

*Figure 5.* Microbenchmark for recall rates of top-32 tokens at $16\times$ sparsity for HashAttention and DS. We can see that HashAttention consistently outperforms DS at varyingn context lengths.

### C.2. Detailed pareto curves for each dataset averaged in the main paper.

The pareto curves are shown in Figure 6

*Table 4.* Longbench (non-chinese tasks)

| | multifieldqa_en | passage-retrievel_en | samsum | qasper | musique | gov_report | multi-news | trec | hotpotqa | passage-count | qmsum | 2wiki | triviaqa | narrativeqa | lcc | repobench-p | AVG | AVG$^{pc}$ |
|---|---|---|---|---|---|---|---|---|---|---|---|---|---|---|---|---|---|---|
| HashAttention-16x | 53.65 | 100 | 42.82 | 43.52 | 29.16 | 34.99 | 26.68 | 70.29 | 51.92 | 1.85 | 24.75 | 42.97 | 92.14 | 27.93 | 64.89 | 60.49 | 48.00 | 51.08 |
| Full | 54.68 | 100 | 43.75 | 45.80 | 29.41 | 34.90 | 27.11 | 71.43 | 55.40 | 8.29 | 25.14 | 45.62 | 90.55 | 30.19 | 62.87 | 55.31 | 48.78 | 51.48 |

*Table 5.* Longbench (chinese tasks). The hash attention is trained on english data only

| multifieldqa_zh | dureader | vcsum | passage-retrieval-zh | lsht | AVG |
|---|---|---|---|---|---|
| 59.31 | 30.93 | 17.48 | 91.71 | 41.71 | 48.228 |
| 61.36 | 34.51 | 17.14 | 96.86 | 44.57 | 50.888 |

### C.3. Latency of SCORE computation

The latency comparison of SCORE computation and full inner product computation is presented in Table 6

*Table 6.* In order to understand the number of bits upto which HashAttention would show latency gains, we compare the full inner product computation latency with bitwise score computation latency for varying number of bits We can see that upto 512 bits, our bit-wise kernel is superior to full inner product computation.

| Latency of SCORE computation | | | | | |
|---|---|---|---|---|---|
| | Inner product | Hamming distance using packed 64 bit integers | | | |
| #tokens | 128 x 16Float | 64bit | 128bit | 256bit | 512bit |
| 262144 | 0.13 | 0.085 | 0.084 | 0.087 | 0.112 |
| 524288 | 0.266 | 0.085 | 0.086 | 0.117 | 0.218 |
| 1048576 | 0.619 | 0.087 | 0.129 | 0.226 | 0.429 |
| 2097152 | 1.698 | 0.144 | 0.247 | 0.445 | 0.85 |
| 4194304 | 3.272 | 0.28 | 0.485 | 0.88 | 1.694 |
| 8388608 | 6.733 | 0.552 | 0.96 | 1.755 | 3.378 |
| 16777216 | 8.306 | 1.1 | 1.911 | 3.504 | 6.746 |
| 33554432 | 16.61 | 2.188 | 3.817 | 7.005 | 13.48 |

*Table 7.* Average Cosine similarity between query and key embeddings of top-32 tokens. This is computed for Llama-3.1-8B-Instruct model

| | Average Cosine Similarity of top-32 tokens | | |
|---|---|---|---|
| Layer number | original embeddings | HashAttention (tanh) | HashAttention(sign) |
| 0 | -0.111818 | 0.1777 | 0.180198 |
| 4 | -0.119951 | 0.4922 | 0.179186 |
| 8 | -0.109657 | 0.2676 | 0.1802 |
| 12 | -0.10329 | 0.1309 | 0.178136 |
| 16 | -0.110954 | 0.2422 | 0.182292 |
| 20 | -0.0985931 | 0.2188 | 0.179659 |
| 24 | -0.114441 | 0.293 | 0.179346 |
| 28 | -0.101307 | 0.2441 | 0.179191 |

## C.4. Does HashAttention resolve the out-of-distribution issue of queries and keys

While HashAttention does not directly aim to resolve the out-of-distribution, the learned mappings do cause the distributions to be closer. We present some data in Table 7

## C.5. Learned bit representations performed in HashAttention vs. data-agnostic bit signatures using LSH (random signed projections) (Gionis et al., 1999)

If we use LSH instead of learned bit signatures via mapping functions as done in HashAttention, the bit signature sizes needed for LSH are considerably higher. This is expected since LSH does not exploit the data distribution in the selection of projections. The results on LongBenchmark datasets is presented in Table 8. This is consistent with results from MagicPig(Chen et al., 2024).

## C.6. Quality vs. bits

The Table 9 shows the cross entropy loss (lower the better) for top-64 prediction while using different bit-widths.

*Table 8.* If we use LSH (used in MagicPig(Chen et al., 2024)) instead of learned bit signatures via mapping functions as done in HashAttention, the bit signature sizes needed for LSH are considerably higher. This is expected since LSH does not exploit the data distribution in the selection of projections

| Compression | | #bits | passage_retrieval_en | multifieldqa_en | hotpotqa | triviaqa | Average |
|---|---|---|---|---|---|---|---|
| 16x | **HashAttention** | 32 | 99.43 | 55.18 | 52.34 | 93.32 | 75.0675 |
| 16x | **LSH** | 32 | 48.57 | 32.62 | 34.38 | 80.02 | 48.8975 |
| | **LSH** | 256 | 89.14 | 47.97 | 53.97 | 88.07 | 69.7875 |
| | **LSH** | 512 | 85.14 | 50.38 | 51.42 | 87.6 | 68.635 |
| | **LSH** | 1024 | 91.43 | 49.75 | 55.25 | OOM | |
| 8x | **LSH** | 32 | 69.14 | 42.71 | 46.63 | 81.78 | 60.065 |
| | **LSH** | 256 | 82.86 | 50.18 | 52.06 | 90.3 | 68.85 |
| | **LSH** | 512 | 95.43 | 52.64 | 54.85 | 87.51 | 72.6075 |
| | **LSH** | 1024 | 98.86 | 51.84 | 54.67 | OOM | |

*Table 9.* Cross-entropy loss for different embedding dimensions evaluated on $250 \times 32K$ samples.

| Dimension | Cross Entropy Loss @ $250 \times 32K$ samples |
|---|---|
| 16 | 0.243 |
| 32 | 0.212 |
| 64 | 0.204 |

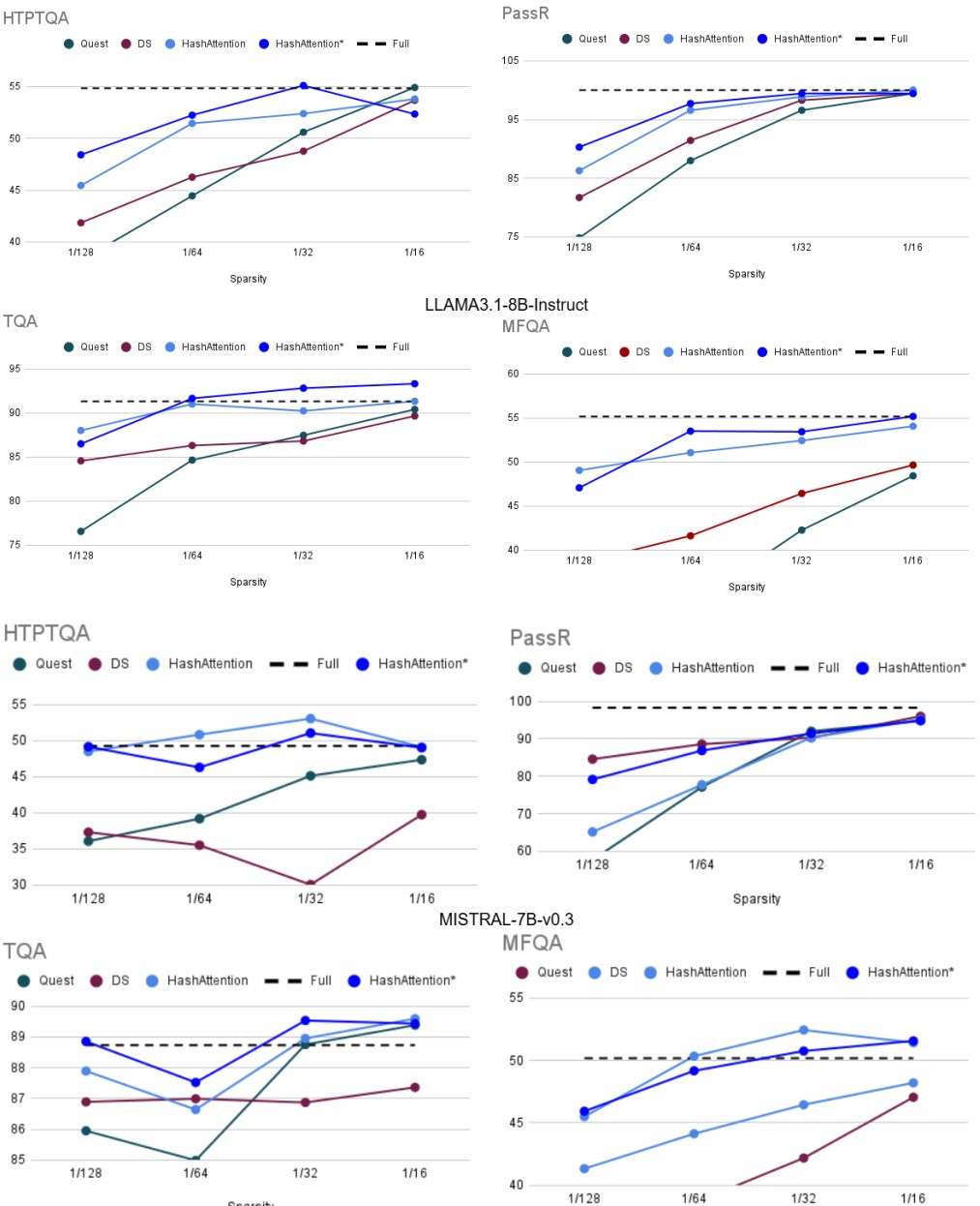

*Figure 6.* Pareto curves at 32 bits PTPA for HashAttention, DS and Quest

