# OpenReview forum: "HashAttention: Semantic Sparsity for Faster Inference"
_ICML.cc/2025/Conference — ICML 2025 poster_

### Official Review · Reviewer_p8mb · 2025-03-08

**Overall Recommendation:** 4

**Summary:**

This paper proposes a simple, effective and plug-and-play method for accelerating inference in autoregressive transformers via top-k attention. The authors propose to accelerate top-k operation by learning mappings to encode queries and keys in Hamming space in a way that the ranking induced using negative Hamming distance of encoded queries and keys follows the ranking induced by original exp(<q, k_i>) * ||v_i||. By efficiently identifying the top keys using bit operations, HashAttention reduces attention computation. Authors demonstrate impressive sparsity levels (up to 32×) with minimal quality degradation, and significant latency improvements (up to 4.3× in GPT-FAST and 2.54× in FlashDecode) with modest auxiliary memory requirements (32 bits per token).

**Claims And Evidence:**

yes

**Essential References Not Discussed:**

none

**Experimental Designs Or Analyses:**

yes

**Methods And Evaluation Criteria:**

yes

**Other Comments Or Suggestions:**

.

**Other Strengths And Weaknesses:**

## Strengths
1. HashAttention consistently outperforms existing sparse attention methods across a wide range of benchmarks. Even when trained on generic data, it achieves 16× sparsity with minimal quality loss on LongBench and RULER benchmarks. With task-specific fine-tuning, sparsity can be pushed to 32× for certain tasks.

2. The use of bit operations for Hamming distance computation bitcount(bitwise_xor(phi(q), phi(k))) is computationally efficient. The approach shows impressive latency improvements in real-world inference systems like GPT-FAST and FlashDecode.

3. At just 32 bits of auxiliary memory per token, HashAttention is significantly more memory-efficient than competitive approaches like Double Sparsity, which require more auxiliary memory to achieve similar quality.

4. The paper presents comprehensive evaluations across multiple datasets, models, and metrics, comparing against a range of state-of-the-art baselines.

## Weaknesses

1. Unlike some heuristic-based approaches, HashAttention requires training on task data. While this enables better performance, it introduces additional complexity for deployment. However, this is a minor point.

2. The paper could more thoroughly discuss how hyperparameters like bits per embedding are selected and their impact on performance. It would help to include code in the appendix.

3. The authors acknowledge that for shorter contexts (< 8K tokens), the overhead of computing bit signatures can outweigh the benefits of sparse attention. This limitation should be more prominently discussed - does this overhead become more prominent with quantized KV cache?

4. The method formulation could be cleaner. For example: z = sigmoid(FF(x)), phi(x) = (z.round() - z).detach() + z.

**Questions For Authors:**

1. Can you include hyperparameters? How much training was required for learning on OpenWebText? Are the LLM weights kept frozen during the learning of hash functions?

2. What's the performance overhead of computing the Hamming distance across all tokens, and at what point does this become a bottleneck compared to full attention computation?

3. Recent papers report that different layers/heads can benefit from different sparsity levels (e.g., middle layers allow higher sparsity). Have you explored variable sparsity across different layers/heads?

4. How does HashAttention perform with quantized KV cache implementations? Does the additional auxiliary memory requirement become more significant in these settings?

**Relation To Broader Scientific Literature:**

The proposed method is sufficiently novel.

**Theoretical Claims:**

yes

---

> ### Author Rebuttal · Authors · 2025-03-31
>
> We thank the reviewer for supporting our paper. Please find responses to the questions and comments below. Kindly let us know if you have any additional follow-up questions.
>
>
> 1. **Hyperparameters for HashAttention Training / Frozen backbone LLM**
>
> 	Yes the LLM weights are kept frozen during training only hash attention mappings are trained locally and independently for each attention head. Our training for 32K length HashAttention on openwebtext looks like
>
> |                            |             |
> |----------------------------|-------------|
> |    Openwebtext samples   |     ~128K    |
> |     32K length samples     |     3750    |
> |    # queries per sample    |     8192    |
> | # total queries in dataset |   30M  |
> |          optimizer         | Adam(0.001) |
>
> 2. **performance overhead of computing the Hamming distance vs. full computation**
>
> Kindly refer to the response 2 (Reviewer HrF1) for the table. We make the following observations,
>
> A. (Sequence dimension) Improvement with using hamming distances over inner products stays consistent with increasing tokens in KV Cache as expected.
>
> B. (Bitwidth dimension) We see that we can go upto as large as 512 bitwidth signatures while maintaining advantageous latency for Hamming distance computation. In our experiments, we use 32 bits for HashAttention.
>
> With such an efficient distance computation, the Hamming distance computation is far from being the bottleneck of HashAttention.
>
> 3. **different sparsity levels for different layers / heads**
>
> We have not explored this dimension. However, we expect such improvements to orthogonally improve all the methods.  We leave this for future work.
>
> 4. **HashAttention performance with quantized KV cache implementations**
>
> Quantization of KV Caches is a way to overall improve the memory footprint of the Cache. Since, in such a setup, HashAttention still would act upon full precision vectors (since we will have access to those while computing signatures), the topk accuracy/sparsity tradeoff would remain the same. Quantization of KV Caches adds another layer of approximation to attention computation. We believe it affects all the sparse attention methods in the same manner.
>
> The absolute memory footprint of HashAttention remains the same. Auxiliary memory would, of course be higher in relative terms. However, it is true for all sparse attention baselines.
>
> **Other discussion:**
>
> 1. **Code**: The anonymous repository for our code used for running long benchmarks is here: https://anonymous.4open.science/r/HashAttention_ICML2025/README.md
> We plan to release full code on acceptance
>
> 2. **Choosing #bits**. In our experiments, we use 32 bits since it works reasonably. As expected, we find that using larger bits helps with the quality of top-k prediction.
>
> | Dimension | Cross Entropy Loss @ 250 x 32K samples |
> |-----------|--------------------|
> |     16    |        0.243       |
> |     32    |        0.212       |
> |     64    |        0.204       |
>
> We will add more results and discussion on these hyperparameters in final version.
>
> 3. **overhead of signatures for < 8K contexts, Does overhead become more prominent with quantized KV**
>
> We will highlight this overhead in the limitation section of the paper for clarity.  We do not believe the overhead increases with quantized KV -- firstly, nothing changes in the signature computation even with quantized KV since we have access to full KV when signatures are created. Secondly, using quantized KV adds dequantize step in the attention computation, which would reduce the relative overhead of signature computation.
>
> 4. **Formulation**: we will improve the formulation in the paper.

---

> > ### Comment · Reviewer_p8mb · 2025-04-02
> >
> > Thank you for the response - I am maintianing my score of Accept.

---

### Official Review · Reviewer_buih · 2025-03-11

**Overall Recommendation:** 4

**Summary:**

Dynamic sparse attention has been widely explored in long-context scenarios. This paper proposes a learned hash function-based token-level dynamic sparse loading method. Specifically, it formulates the sparse attention top-K problem as a recommendation task, utilizing a learnable hash function to predict top-K tokens.
The learned hash function is implemented as a three-layer MLP, generating a 32-bit hash score. During decoding, the method computes the query hash score, selects top-K tokens, and performs token-level gather sparse attention.
The approach is evaluated on LLaMA-3.1-8B and Mistral-v0.3-7B using LongBench and RULER, with end-to-end and kernel-level latency comparisons against FlashDecoding and GPT-Fast. Results show that in long-context scenarios, the proposed method achieves up to 4.3× speedup over GPT-Fast.

**Claims And Evidence:**

The paper's claims are fairly accurate, as it is the first work to leverage a learnable hash function to address the sparse attention top-K problem.

**Essential References Not Discussed:**

MagicPIG, a concurrent study, also utilizes LSH for sparse attention. It would be beneficial to include a discussion comparing the advantages and limitations of these two approaches. Could you provide a discussion comparing the advantages and limitations of these two approaches?

**Experimental Designs Or Analyses:**

1. The paper lacks a comparison with training-free hash function methods, such as MagicPIG[1], which also employs LSH for sparse attention. Evaluating against such baselines would help clarify the advantages and limitations of the proposed approach.
2. Additionally, the token-level approach may not be GPU-friendly, yet there is no latency breakdown provided for the gather sparse attention stage. An ablation study on the performance gain from token-level hash retrieval is also missing.
3. The study lacks OOD experiments, as it is trained on a retrieval dataset but is not tested on reasoning or other domain-specific benchmarks. Evaluating generalization across different domains would strengthen the analysis.
4. Finally, there is no discussion or analysis on whether the learned hash index helps mitigate query-key distribution shift issues in sparse attention top-K retrieval. Would you be able to provide additional experiments or insights on this aspect?

[1] MagicPIG: LSH Sampling for Efficient LLM Generation. ICLR 2025.

**Methods And Evaluation Criteria:**

The method design is reasonable. Unlike Quest-based approaches, which rely on block-wise min-max values as centroids, this method learns a hash index, which is likely to provide better representations and mitigate OOD issues. The main concern is the generalization capability of the learned hash index.

Additionally, the choice of a hash index is justified—the hashing process is GPU-friendly, and storing hash results incurs minimal cost.

**Other Comments Or Suggestions:**

Additionally, I noticed an issue in #43, where 128GB should be corrected to 32GB, where LLaMA-3.1-8B uses GQA with a group number of 8.

**Other Strengths And Weaknesses:**

N/A

**Questions For Authors:**

1. Do you have an ablation study using only a training-free hash index?
2. Do you provide a latency breakdown for gather sparse attention?
3. Have you conducted any OOD analysis between the training and test domains?
4. Have you analyzed whether the learned hash index mitigates OOD issues in sparse attention top-K retrieval?

**Relation To Broader Scientific Literature:**

Prior work has explored sparse attention methods for long-context LLMs, leveraging attention sparsity through approaches such as StreamingLLM and H2O. More recently, RetrievalAttention has introduced vector retrieval to optimize sparse attention computation. Building on these efforts, this paper proposes a learnable hash index to make sparse attention top-K selection more GPU-friendly.

**Theoretical Claims:**

The paper also provides an error analysis for sparse attention during decoding and reformulates the MIPS problem using cos-sin transformations with an addition operation. However, this Lemma 4.2 has already been widely adopted in the vector retrieval community.

---

> ### Author Rebuttal · Authors · 2025-03-31
>
> We thank the reviewer for supporting our paper. Please find responses to the questions and comments below. Kindly let us know if you have any additional follow up questions.
> 1. **training-free hashing**
> To compare HashAttention to training free methods, we measure quality of sparse attention while using LSH signatures (random signed projections) in place of learned mapping-based HashAttention signatures.  As expected, the data-agnostic LSH needs longer signatures to start giving reasonable quality.
>
> ||| bits | passage_retrieval_en| multifieldqa_en| hotpotqa|triviaqa|Avg|
> |-----|---------------|:----:|:---------------------:|:---------------:|:--------:|:--------:|:-------:|
> | 16x | HashAttention |  32  |99.43 |55.18      |   52.34  |   93.32  | 75.0675 |
> | 16x |      LSH      |  32  |48.57 |32.62|   34.38  |   80.02  | 48.8975 |
> |     |      LSH      |  256 | 89.14  |47.97|   53.97  |   88.07  | 69.7875 |
> |     |      LSH      |  512 | 85.14 |50.38|   51.42  |   87.6   |  68.635 |
> |     |      LSH      | 1024 |91.43 |49.75|   55.25  |    OOM   ||
> |  8x |      LSH      |  32  |69.14| 42.71|   46.63  |   81.78  |  60.065 |
> |     |      LSH      |  256 | 82.86 | 50.18|   52.06  |   90.3   |  68.85  |
> |     |      LSH      |  512 | 95.43 | 52.64|   54.85  |   87.51  | 72.6075 |
> |     |      LSH      | 1024 | 98.86 | 51.84|   54.67  |    OOM   ||
>
> 2. **Token level hashing / gathering in sparse attention / performance efficiency.**
>
> Our implementation does not involve gathering pivotal tokens into a contiguous memory space. The implementation is built upon the vLLM page attention framework, where each token corresponds exactly to one page (i.e., page size = 1). After identifying top-k tokens, we use their indices without physically moving or gathering these tokens into a separate memory buffer. The page attention kernel explicitly utilizes these indices to selectively compute attention only for the specified tokens, efficiently ignoring irrelevant tokens. The page attention kernel is highly optimized for GPU memory access patterns. Due to the GPU cache line size of 128 bytes, optimal memory bandwidth utilization is achieved as long as contiguous data access meets or exceeds this cache line size. Each token's head representation has 128 fp16 elements, equivalent to 256 bytes. This naturally exceeds the GPU cache line size, allowing our attention kernel to leverage GPU memory bandwidth effectively.
>
> Many state-of-the art inference framework implement attention using paged-attention backbone with page size 1. They find that with correct optimization, the efficiency does not depend on page size. Quoting from official flashinfer documentation
> *“Some recent work such as LightLLM and sglang uses a special form of PageAttention where page size equals one, for easy management of KV-Cache in complicated serving scenarios such as structured generation. FlashInfer optimizes PageAttention kernels by pre-fetching page indices in GPU shared memory, so that kernel performance is not affected by the page size.”*  We will explicitly clarify this and use better naming for the different stages in the paper to avoid potential confusion.
>
> 3. **OOD test sets / generalization of learned mappings across tasks.**
>
>  We show two sets of results in our experiments. The HashAttention is trained on completely unrelated data from openwebtext dataset and tested on OOD LongBench and RULER. The good performance of HashAttention on these datasets is a testament to its strong generalization performance.  Additionally, finetuning for specific benchmark, which gives us HashAttention*, further improves the quality of results.
>
> 4. **OOD Query** Kindly refer to Response 4 to reviewer HrF1.
>
> 5.  **MagicPig vs. HashAttention:**
>
> MagicPig can be understood and compared with HashAttention in two respects.  Firstly identifying important tokens. HashAttention uses succinct bit signatures (32 bits) obtained via learned mappings to compute important tokens. MagicPIG uses data-agnostic LSH to obtain bit signatures. As expected MagicPIG needs much longer bit signatures to obtain reasonable results ( from their paper 1500-2200 bits). MagicPig builds index on this signatures and due to irregular bucket sizes, these indices need to be stored on CPUs -- a reasonable solution in case of KVCache being stored on CPUs.
>
> Secondly, MagicPig proposes sampling based estimation instead of top-k estimation. Sampling is performed using LSH tables. The idea sampling based estimation is an interesting direction to be explored w.r.t its application to other sparse attention methods, including HashAttention.
>
> We plan to thoroughly compare HashAttention with many concurrent works such as MagicPig, PQCache, Squeeze Attention in our future work. Comparison against these is out of the scope of the rebuttal.
>
> 6. **Lemma 4.2** : We would be happy to cite the correct reference in prior literature. Kindly direct us to the same.
>
> Please let us know if there are any additional queries.

---

> > ### Comment · Reviewer_buih · 2025-04-02
> >
> > Thank you for your detailed response. I have no further questions and maintain my recommendation for acceptance of this paper.

---

### Official Review · Reviewer_HrF1 · 2025-03-13

**Overall Recommendation:** 4

**Summary:**

The authors propose a method to identify relevant tokens in the attention computation by framing it as a MIPS search, using the relationship between MIPs and cosine similarity plus the approximation of cosine similarity in terms of the hamming distance of the corresponding hamming embeddings. The authors propose a framework, which other sparse attention approaches can be fit into, which consists on a scoring, topk and gather attention steps. Hash attention is first trained offline with generic data. The authors learn independent mappings for key-value pairs and queries and then use the hamming distance between these mappings to identify the pivotal tokens.

**Claims And Evidence:**

The claims made in the submission supported by clear and convincing evidence.

**Essential References Not Discussed:**

They cite up to date papers such as Retrieval Attention, Magic Pig and Squeezed attention, among others.

**Experimental Designs Or Analyses:**

Yes, table 1-2 and figure 3

**Methods And Evaluation Criteria:**

The methods and/or evaluation criteria chosen make sense for the problem or application at hand.

**Other Comments Or Suggestions:**

It is not clear why retrieval augmented generation approaches are discussed in related work but these are not compared against in the experiments section. Even though RAG is a competitive approach to reduce the context length, it doesn't immediately relate to dynamic sparsity attention computation so I suggest to remove this from the related work section.

**Other Strengths And Weaknesses:**

The paper includes a variety of experiments and evaluations that showcase the different strengths of the method.

**Questions For Authors:**

What is the dimension of the hamming codes to guarantee that it is indeed cheaper than computing the cosine similarity?
How do you solve the unbalanced buckets issue in LSH, given that you use a single hash table?
Could you provide further details of how the learning of independent mappings for key-value pairs and queries remedies the OOD problem?
 Would it be feasible to include the topk with k-means approach in the baselines to understand how it does wrt LSH partitioning?
Could you provide further details about the extra fine-tuning on datasets for the experiments?

**Relation To Broader Scientific Literature:**

The authors include a wide overview for the works in the area and classify them in fixed and dynamic sparsity plus token eviction methods.

**Theoretical Claims:**

Lemma 4.2 presents a true score for token relevance that has the same ordering as using the inner product between queries and key-value pairs plus states the equivalent with respect to cosine similarity. These lemma includes a proof in the appendix and the relevant references to support it.

---

> ### Author Rebuttal · Authors · 2025-03-31
>
> We thank the reviewer for supporting our paper. Please find responses to the questions and comments below. Kindly let us know if you have any additional follow up questions.
>
> 1. **Discussion on RAG** – We will remove this from related work.
>
> 2. **Dimension of hamming codes:**
>
> We can answer this question by comparing (1) the  latency of inner product computation ( dim=128 which is the standard dimension in most models)  and (2) the hamming distance computation based on our kernel implementation.
>
> | Latency of SCORE computation ||||||
> |:----------------------------:|:-------------:|:---------------------------------------------:|:------:|:------:|:------:|
> |                              | Inner product | Hamming distance||||
> |            #tokens           | 128 x 16Float |                     64bit                     | 128bit | 256bit | 512bit |
> |            262144            |      0.13     | 0.085                     |  0.084 |  0.087 |  0.112 |
> |            524288            |     0.266     | 0.085                     |  0.086 |  0.117 |  0.218 |
> |            1048576           |     0.619     |0.087                     |  0.129 |  0.226 |  0.429 |
> |            2097152           |     1.698     |0.144                     |  0.247 |  0.445 |  0.85  |
> |            4194304           |     3.272     |0.28                     |  0.485 |  0.88  |  1.694 |
> |            8388608           |     6.733     |0.552                     |  0.96  |  1.755 |  3.378 |
> |           16777216           |     8.306     |1.1                      |  1.911 |  3.504 |  6.746 |
> |           33554432           |     16.61     |2.188                     |  3.817 |  7.005 |  13.48 |
>
> We can see that up to 512 bits, the hamming distance is cheaper than the inner product computation. In practice, we do not need such large bit widths. This excludes the cost of running MLP on a query vector, which does not scale with #tokens.
>
> 3. **unbalanced buckets issue in LSH**
>
> Since HashAttention compares the query signature with all token signatures, it is not directly affected by the unbalanced bucket issue. However, it can affect top-k selection when many tokens share the same Hamming distance. In such cases, we randomly chose $k$ tokens in our implementation.
>
> 4. **query OOD issue**
> The mappings learned in hashattention transform query and key-value into a low-dimensional semantic space where the smaller the hamming distance, better is the relevance of key-value to the query. While training the mappings, this hamming distance is used to classify the key-value to be relevant or irrelevant to the query. The setup naturally promotes transformed query distribution to be closer to key-value distribution, as shown in the table below. We can see that average cosine similarity significantly improves after the transformation. We provide results for both soft embeddings and hard embeddings for HashAttention.
>
>
> |              | Average Cosine Similarity of top-32 tokens |                      |                     |
> |--------------|:------------------------------------------:|:--------------------:|:-------------------:|
> | Layer number | original embeddings                        | HashAttention (tanh) | HashAttention(sign) |
> |            0 |                                  -0.111818 |               0.1777 |            0.180198 |
> |            4 |                                  -0.119951 |               0.4922 |            0.179186 |
> |            8 |                                  -0.109657 |               0.2676 |              0.1802 |
> |           16 |                                  -0.110954 |               0.2422 |            0.182292 |
> |           24 |                                  -0.114441 |                0.293 |            0.179346 |
>
> 5. **K-means based topk vs. vanilla LSH vs. HashAttention:**
>
> HashAttention uses learned mappings to obtain bit-signatures for retrieval (while motivated by LSH, it does not perform LSH). Due to rising interest in this topic, many concurrent works have explored different approaches from information retrieval. These include MagicPig (which uses LSH tables), SqueezeAttention and PQCache (which use clustering), and RetrievalAttention (which employs graph-based retrieval). We plan to thoroughly compare HashAttention with these works in future research. However, such a comparison is beyond the scope of this rebuttal
>
> 6. **Extra fine-tuning details**
>
> To fine-tune HashAttention mappings for downstream LongBench, we use 25 samples from all LongBench task each to create a fine-tuning dataset. Then we further train hashAttention on this dataset. The evaluation is performed on the LongBench excluding samples included in the fine-tuning dataset.

---

> > ### Comment · Reviewer_HrF1 · 2025-04-08
> >
> > Thank you for your detailed explanations, I encourage you to add these to the appendix in the camera ready version if possible.

---

### Official Review · Reviewer_fb4H · 2025-03-17

**Overall Recommendation:** 4

**Summary:**

This paper introduces HashAttention, framing pivotal token identification as a recommendation problem. Given a query, HashAttention encodes keys and queries in Hamming space, capturing the required semantic similarity, using learned mapping functions. HashAttention efficiently identifies pivotal tokens for a given query using bitwise operations and computes attention using only these tokens, improving the overall attention efficiency. Trained on generic data, HashAttention reduces tokens used by up to 16× with minimal quality loss, requiring only 32 bits of auxiliary memory per token.

**Claims And Evidence:**

From the theoretical analysis, HashAttention transforms the problem into a maximum inner product search problem and approximates it in Hamming space by learning mapping functions, which has a reasonable theoretical. The experimental results support the effectiveness. On multiple data sets and models, HashAttention performs better than baselines in the same auxiliary memory budget.

**Essential References Not Discussed:**

Not yet.

**Experimental Designs Or Analyses:**

The experimental design was comprehensive, comparing multiple baselines, covering different models and data sets, and considering different assessment indicators, such as quality, efficiency, recall rate.

**Methods And Evaluation Criteria:**

Compared with the existing methods, progress has been made in reducing the use of KV cache and improving the efficiency, and improving the efficiency of attention computation.

**Other Comments Or Suggestions:**

No.

**Other Strengths And Weaknesses:**

Strengths
1.The idea of paper is interesting, e.g., transforming key token recognition into recommendation problem and realize efficient attention calculation by coding in Hamming space.
2.Well-written paper with a clear process, the English writing is easy to follow.
3.Extensive experiments demonstrate the validity of the model.

Limitations
1.Supplement the experiment in long context where KV cache is located in CPU RAM, and compare the performance of HashAttention with other baselines.
2.Expand the experiments of HashAttention on complex inference and multimodal tasks.
3.The formula derivation process is elaborated, and the key terms are explained in detail.

**Questions For Authors:**

No.

**Relation To Broader Scientific Literature:**

HashAttention transforms the problem into a maximum inner product search problem and approximates it in Hamming space by learning mapping functions, which has a reasonable theoretical and good application value.

**Theoretical Claims:**

From the theoretical analysis, HashAttention transforms the problem into a maximum inner product search problem and approximates it in Hamming space by learning mapping functions, which has a reasonable theoretical.

---

> ### Author Rebuttal · Authors · 2025-03-31
>
> We thank the reviewer for supporting our paper. We are working on extending Hash Attention to scenarios involving KV cache offloading and reasoning tasks, and we will present these in our future work. Please let us know if you have any other questions; we would be happy to clarify.

---

### Decision · Program_Chairs · 2025-05-01

**Decision:**

Accept (poster)

**Comment:**

The paper presents a method for long-context attention. It is trained to learn an embedding of the key-values and queries into Hamming space. It then imposes sparsity on the attention coefficients by searching for relevant tokens with a max-inner-product search over the Hamming embeddings and attending only to the top-k search results. The method's design enables a GPU-friendly implementation. Intuition is drawn from cosine similarity search with LSH.


Reviewers were unanimously positive about the comprehensive experiments and empirical improvement. The rebuttals contained useful expansions and empirical results which the authors are encouraged to include in the final version.